# The Use of Magnetic Resonance Imaging Techniques in Assessing the Effects of Alcohol Consumption and Heavy Drinking on the Adolescent Brain: A Scoping Review Protocol

**DOI:** 10.3390/brainsci11060764

**Published:** 2021-06-09

**Authors:** Nancy Hornsby, Soraya Seedat, Eric Westman, Lars-Olof Wahlund, Nandi Siegfried, Lesley-Ann Erasmus-Claassen, Bronwyn Myers

**Affiliations:** 1Alcohol, Tobacco and Other Drug Research Unit, South African Medical Research Council, Cape Town 7505, South Africa; Nandi.Siegfried@mrc.ac.za (N.S.); Lesley-Ann.Erasmus@mrc.ac.za (L.-A.E.-C.); Bronwyn.Myers@mrc.ac.za (B.M.); 2Department of Psychiatry, Stellenbosch University, Cape Town 7505, South Africa; sseedat@sun.ac.za; 3Division of Clinical Geriatrics, Department of Neurobiology, Care Sciences and Society, Karolinska Institutet, 141 83 Stockholm, Sweden; eric.westman@ki.se (E.W.); lars-olof.wahlund@ki.se (L.-O.W.); 4Division of Addiction Psychiatry, Department of Psychiatry and Mental Health, University of Cape Town, Cape Town 7935, South Africa

**Keywords:** scoping review protocol, adolescent alcohol use, magnetic resonance imaging

## Abstract

*Introduction*: Alcohol consumption, specifically heavy drinking during adolescence, has been shown to be accompanied by adverse structural brain changes in adolescent drinkers. This scoping review will aim to quantify and evaluate the quality of studies in which magnetic resonance imaging (MRI) techniques are used to assess regional brain deficits among adolescents who consume alcohol. *Methods and analysis*: This scoping review will be conducted following the Arksey and O’Malley scoping review methodology framework and will be reported using Preferred Reporting Items for Systematic Reviews and Meta-Analyses (PRISMA) extension for Scoping Reviews (PRISMA-ScR) guidelines. Literature will be searched for the period January 1999 to March 2021. Two reviewers will independently screen titles/abstracts and full-texts in two consecutive screening stages. Eligible studies will be independently reviewed to ensure that inclusion criteria are met. Cohen’s Kappa (κ) will be used to calculate inter-rater agreement. A third reviewer will resolve any disagreements. The Joanna Briggs Institute (JBI) Appraisal Tools will be used for quality appraisal of the included studies. Findings will be reported by means of a narrative overview, tabular presentation of study characteristics, and quality assessment, and a thematic analysis of major themes. This scoping review has been registered with the Open Science Framework. *Ethics and dissemination*: Scoping reviews do not require ethical approval, however, this review forms part of a larger study that has obtained approval from the Faculty of Health and Medical Sciences, Health Research Ethics Committee at Stellenbosch University (S20/04/086). Findings will be disseminated by means of peer-reviewed publications and conferences.

## 1. Strengths and Limitations of This Study

This scoping review provides a critical review of the MRI techniques in identifying neuroanatomical brain deficits associated with adolescent alcohol consumption.In addition to the mapping of the evidence, this scoping review will provide a critical appraisal of the evidence using appropriate Joanna Briggs Appraisal Tools.A comprehensive search strategy was developed with the assistance of an information specialist and a public health systematic review expert. In addition, the technical aspects of the methodology was guided by the public health systematic review expert.The team consisted of experts in addiction research, neuroimaging techniques, and public health systematic reviews.This scoping review was registered with the Open Science Framework (https://osf.io/n5xud/ accessed on: 20 March 2020).

## 2. Introduction

Substance use disorders including alcohol and other drug use disorders continue to be a major public health concern and a large contributor to mortality and morbidity worldwide [1]. According to the Global Burden of Disease Study (2019), alcohol was among the top ten leading risk factors across all ages, contributing to 3.7% (3.3–4.1) of global disability adjusted life years (DALYs) in 2019 [2]. Among individuals aged 10 to 24 years, alcohol has become the second leading risk factor, contributing to 2.6% (2.1–3.1) of DALYs [2]. Additionally, in 2019, alcohol accounted for 0.374 million (0.298–0.461) deaths among females and 2.07 million (1.79–2.37) deaths among males [2]. Alcohol-related morbidity has been well-established in adolescents and is associated with heightened risk for suicide, HIV, and other sexually transmitted diseases, interpersonal violence, and traffic-related injuries, among others [3,4,5,6]. Potential links are thought to exist between heavy episodic drinking (HED) during adolescence and lifetime alcohol use disorder (AUD) [1,7].

With an estimated 26.5% (or 155 million) of adolescents aged 15–19 years reporting current use of alcohol [3], 45.7% of whom engage in HED, it is not surprising that alcohol-related research has been identified as one of the top research priorities for the promotion of adolescent health [8,9].

### 2.1. Adolescence as a Critical Period for Brain Development

Heavy drinking is a particular concern during adolescence as this is a critical developmental period marked by differential maturation rates of cortical and subcortical brain structures. Evidence suggests a neuroanatomical basis for adolescent alcohol use. Mesolimbic structures responsible for reward and prefrontal cortical (PFC) structures responsible for inhibitory control have been implicated in the risk for adolescent alcohol consumption [10,11,12]. Significant neuronal changes that involve pruning of inactive synapses, development of new synaptic connections, formation of myelin sheaths around the axons (myelinogenesis), and changes in concentrations of neurotransmitters and their respective receptor levels occur during this period [7,13,14]. Incomplete neuromaturation of various brain regions, specifically, the prefrontal cortex (PFC), enhances vulnerability to external influences such as trauma and the effects of alcohol and other substances [13]. Adolescents may experience deficits in cognitive (executive functioning, attention, memory), psychological (depression, anxiety, stress), and behavioral (aggression, injury, violence) functioning [15,16]. While neuronal changes occur in many areas of the adolescent brain, the prefrontal cortex (PFC) undergoes protracted structural and functional maturation into the early twenties [13,17]. In contrast to the later maturation of the PFC, the mesolimbic system and the ventral striatum in the basal ganglia mature earlier and faster, leading to an imbalance between inhibitory control (PFC) and reward (mesolimbic system and ventral striatum) and an overriding control by the desire for reward and overriding control of reward mechanisms [18,19].

The onset of alcohol consumption and risky drinking behaviors usually occur during this critical neurodevelopmental period when PFC structures are not yet fully matured, rendering the adolescent brain especially vulnerable to structural and functional disruption as a result of alcohol use [17,20]. Heavy drinking, particularly heavy drinking that occurs during concentrated periods of time (binges), is associated with extensive neuromorphological disruptions in the adolescent brain [21,22]. Alcohol exposure together with immaturity of PFC structures and vulnerable neurophysiology contribute to an increased risk of cognitive deficits in adolescents.

A number of brain structural (sMRI) and functional (fMRI) magnetic resonance imaging (MRI) studies have been conducted among adolescents who use alcohol in high income settings. These studies have identified the anterior cingulate cortex (ACC), situated in the medial aspect of the cortex, and the ventral striatum, a principal component of the basal ganglia, as key neuroanatomical structures for reward processing that serve to integrate “predicted cost and reward in a net value signal” [12]. Though there is a growing body of research using MRI techniques to elucidate the impact of alcohol consumption on the adolescent brain, there is a paucity of reviews synthesizing the evidence that could be informative for future research in the area. To the authors’ knowledge, the most recent reviews of brain MRI studies for adolescent alcohol use were undertaken in 2014 [23] and 2017 [24]. The 2014 review included 21 studies (10 sMRI and 11 fMRI) covering the period 2000 to 2014 and provided a qualitative synthesis of the evidence without appraising the quality of the studies. Similarly, the 2017 review provides an overview of structural and functional abnormalities associated with heavy drinking during adolescence without providing an assessment of the quality of the evidence included. Brain MRI investigation of adolescent alcohol use has grown substantially since 2017, necessitating an updated review and synthesis of new evidence. This review seeks to update the previously published review and provide a critical appraisal of the quality of the studies included, which the previous reviews did not offer. While it is important and useful to generate a summary of the scientific literature, it is also imperative that the scientific robustness of the evidence be investigated, which the current review will do. In addition, the 2017 review was not a systematic/scoping review and it is therefore possible that relevant literature could have been missed.

### 2.2. Study Aims and Objectives

The aim of this systematic scoping review is to identify the quantity and evaluate the quality of studies examining brain magnetic resonance imaging (MRI) studies (including sMRI, fMRI, diffusion tensor imaging, and susceptibility weighted imaging) investigating neuroanatomical deficits associated with adolescent alcohol consumption including heavy episodic drinking (HED). Stated otherwise, this review will map the evidence rather than determining the effectiveness or diagnostic test accuracy of MRI techniques in the field of adolescent alcohol use.

## 3. Methodology

A scoping review using systematic methodology will be conducted to appraise extant literature and to identify gaps in current research on the use of MRI techniques in adolescent alcohol use research. The review will not be limited to age to ensure inclusion of studies with participants older than 18 years where adolescents are analyzed as a distinct subgroup. The review will also not be limited by study design. The scoping review will follow the Arksey and O’Malley [25] scoping review methodology framework and will be reported using the Preferred Reporting Items for Systematic Reviews and Meta-Analyses [PRISMA] extension for Scoping Reviews (PRISMA-ScR) [26]. The methodological approach by Arksey and O’Malley follows six stages: (i) identifying the research question; (ii) identifying and searching relevant studies; (iii) study selection; (iv) data extraction; (v) collating, summarizing and reporting the results; and (vi) an optional step that involves consulting stakeholders for validation of findings [25,27]. This protocol has been registered with the Open Science Framework (OSF, https://osf.io/n5xud/ access on: 20 March 2020).

### 3.1. Stage 1: Identifying the Research Question

The research question emanates from a larger study investigating the neuroimaging, neurocognitive, and neurohormonal risk markers of heavy alcohol consumption during adolescence. The literature on MRI and adolescent alcohol consumption is extensive, however, synthesis of the evidence is limited and dated. In addition, no review could be found that presents a critical appraisal of the scientific robustness of MRI studies.

In developing the research question, the first author used the Population, Intervention/Exposure, Comparison, Outcome (PICO) Model as guidance. The research question for the proposed review was broadly defined to allow for a wide review of neuroimaging literature around adolescent alcohol use and heavy alcohol consumption:

What is the extent and the quality of the evidence for the use of magnetic resonance neuroimaging techniques in assessing neuroanatomical deficits associated with alcohol consumption and heavy episodic drinking (HED) during adolescence?Specific questions were:*(a)* *To what extent have magnetic resonance imaging techniques been used to assess regional brain region associated with adolescent alcohol use and HED? and;**(b)* *What is the strength of the evidence on magnetic resonance imaging of regional brain deficits in adolescents with alcohol use and HED?*

### 3.2. Stage 2: Searching and Identifying Relevant Studies

A comprehensive search strategy was developed to answer the research question. The first author consulted with co-authors and an experienced information specialist at the South African Medical Research Council’s (SAMRC) Cochrane South Africa unit to develop a search strategy. Decisions were made about search databases, time frames, language, search terms, and syntax strategies to ensure that these were inclusive enough to capture relevant studies.

## 4. Search Strategy and Data Sources

We will conduct an English-only language search of databases for the period January 1999 to March 2021. This period was selected as neuroimaging and adolescent alcohol consumption studies only started to emerge in the early 2000s. Search databases included the Cochrane Library, PubMed, and Scopus. The search will be supplemented by means of reference list searches and grey literature (i.e., theses, conference presentations/abstracts). Trial registers will also be consulted to expand the search. Additionally, texts from case studies, reviews, theses, letters to editors, editorials, commentaries, and conference abstracts will be reviewed for full text published research. Study authors will be contacted in cases where texts cannot be sourced.

### 4.1. Search Strategy

Eligibility criteria and search terms were defined by the first author in consultation with co-authors and the information specialist (IS). The IS verified search terms as well as additional search terms that came up during an initial test search. After this step, the search strategy was further developed to include Medical Subject Headings (MeSH) terms, Boolean logic and operators (‘and’, ‘or’, ‘not’), and filters to ensure accuracy of searches across the different databases. MeSH terms and keywords from previous systematic reviews were also incorporated. The terms will also be searched in the title and abstract (tiab) field code. This strategy resulted in the inclusion of the search terms in Table 1.

### 4.2. Selection Criteria

Articles will be included in the review if they meet the following inclusion criteria.

#### 4.2.1. Inclusion Criteria

(a)Studies in all ages where the study sample, or a distinct subgroup of the sample, is comprised of adolescents aged 13 to 18 years;(b)Studies using MRI techniques;(c)Studies investigating current alcohol use (one drink in the past 30 days) and/or heavy consumption (defined as eight or more drinks per week for females and 15 or more drinks per week for males) and/or heavy episodic drinking (defined as the consumption of five or more drinks for males and four or more drinks for females during a single occasion) [27];(d)Studies investigating polysubstance use only if alcohol is the primary focus of the investigation;(e)Studies including other imaging modalities only if MRI sequences (sMRI, fMRI, diffusion tensor imaging [DTI]) are included;(f)Published between January 1999 and March 2021;(g)Observational (cross-sectional, case-control, cohort, longitudinal) study designs, meta-reviews, systematic reviews;(h)English-only publications; and(i)Grey literature (theses, conference presentations, and abstracts)

#### 4.2.2. Exclusion Criteria

(a)Studies where the study sample only includes individuals younger than 13 or adults older than 18 years, or where adolescents (13–18) did not form a distinct subgroup of the overall sample;(b)Studies involving neuroimaging techniques other than MRI; and(c)Case studies, letters to editors, editorials, and commentaries.

### 4.3. Stage 3: Study Selection

Studies will be independently screened and reviewed by two authors, (NH, LE-C, and BM) across a two-stage process: stage one will involve title and abstract screening and stage two will involve full-text review and critical appraisal. Both stages will be driven by a collaborative process. Titles and abstracts will be screened against eligibility criteria that have been developed a priori. Articles deemed relevant will be included in the next stage of full-text review. In cases where authors are uncertain of relevance, articles will also be included for full-text review. Full text articles will then be independently reviewed by the two reviewers and assessed against the eligibility criteria. Inter-rater agreement will be calculated by means of Cohen’s Kappa (κ). Disagreements will be resolved through discussions and a third rater, who will be a senior member of the research team.

### 4.4. Quality Appraisal

Quality of included studies will be assessed using the Joanna Briggs Institute (JBI) Critical Appraisal Tools, which allow for the appraisal of the trustworthiness, relevance, and results of the research article [28]. JBI Critical Appraisal Tools have been developed by JBI and their collaborators and approved by the JBI Scientific Committee after an extensive peer-review process. The tools allow for the appraisal of the methodological quality of a research study as well as the risk for bias in the study design, conduct, and analysis [29]. The Checklist for Analytical Cross Sectional Studies (Appendix A) and the Checklist for Cohort Studies (Appendix B) [29] will be used for the appraisal of included studies.

The Checklist for Analytical Cross Sectional Studies appraises studies across eight domains: (1) whether study inclusion criteria were clearly defined; (2) whether the study subjects and the setting were described in detail; (3) whether the exposure was measured in a valid and reliable way; (4) whether objective, standard criteria were used for measurement of the condition; (5) whether confounding factors were identified; (6) whether strategies to deal with confounding factors were clearly stated; (7) whether the outcomes were measured in a valid and reliable way; and (8) whether appropriate statistical analyses were used [28].

The Checklist for Cohort Studies appraises studies across 11 domains: (1) whether the two groups were similar and recruited from the same population; (2) whether the exposures were measured similarly to assign people to both exposed and unexposed groups; (3) whether the exposure was measured in a valid and reliable way; (4) whether confounding factors were identified; (5) whether strategies to deal with confounding factors were clearly stated; (6) whether the groups/participants were free of the outcome at the start of the study (or at the moment of exposure); (7) whether the outcomes were measured in a valid and reliable way; (8) whether the follow-up time was reported and sufficient to be long enough for outcomes to occur; (9) whether follow-up was complete, and if not, whether the reasons for loss to follow up were described and explored; (10) whether strategies to address incomplete follow up were utilized; and (11) whether appropriate statistical analyses were applied. Answers to both tools are either yes, no, unclear or not applicable [28].

The number of studies identified, the number that are included and excluded as well as the reasons for exclusion have been graphically presented in the PRISMA flow diagram (see Appendix C). The flow diagram will present information flow throughout the review stages, mapping out the number of records identified, number of additional records identified, duplicate removal, included and excluded articles, and reasons for exclusion [30]. Included studies will be narratively described.

### 4.5. Stage 4: Data Extraction

Data will be extracted independently by two reviewers (NH, BM) and charted in a metadata MS Excel File. Relevant study details will be methodically captured across the following domains: (a) author(s); (b) country; (c) type of MRI scanner; (d) type of MRI sequence; (e) MRI sequence parameters; (f) image preprocessing and segmentation procedures; (g) statistical software used; (h) study design; (i) study population; (j) outcome measures; (k) main findings; (l) limitations; and (m) recommendations for future research.

### 4.6. Stage 5: Collating, Summarizing, and Reporting of the Results

Findings from the included studies will be presented by means of a descriptive overview. This overview will be accompanied by a table describing the characteristics of included studies, a quality assessment table, and thematic analysis of results. First, collation and summarizing of results will involve the grouping of studies by the MRI technique. Second, the studies and main findings will be synthesized across the different MRI groupings. Finally, a table of strengths and gaps in the evidence [27] will be presented and opportunities for research using MRI in adolescent alcohol use research will be reported.

### 4.7. Stage 6: Consulting Stakeholders for Validation of Findings

This final step of consulting stakeholders is not required for a scoping review [25], however, including this step is deemed as a necessary step that enhances the methodological rigor of a scoping review [27]. MRI experts will be consulted to provide review and feedback on the technical aspects and reporting of MRI research. This will ensure that reporting of MRI technical properties and findings are done accurately. Stakeholders will also be involved in the interpretation stage to ensure that the interpretation of findings is accurate.

## 5. Patient and Public Involvement

There will be no patient and public involvement in the development of this scoping review protocol.

## 6. Ethics and Dissemination

Though scoping reviews do not require ethical approval, the study of which this scoping review forms part of has been approved by the Faculty of Health and Medical Sciences, Health Research Ethics Committee at Stellenbosch University (HREC approval number: S20/04/086). The findings from the proposed review will be disseminated through peer-reviewed publications and conference presentations.

## 7. Conclusions

This scoping review will provide a comprehensive review and critical appraisal of the existing evidence base for the use of MRI techniques in adolescent alcohol use, specifically heavy drinking. Furthermore, it will set out to identify a gap in current research and provide a basis for further neuroimaging research in the field of adolescent alcohol use.

## 8. Strengths

There are several strengths of this review. It will provide a comprehensive, well-structured overview and will include a critical appraisal of the methodological quality of the evidence for sMRI and fMRI techniques in identifying neuroanatomical brain deficits associated with adolescent alcohol consumption. A comprehensive search strategy will include three electronic databases augmented with screening reference lists and grey literature.

## 9. Limitations

The search will be limited to English language only, which means that potential eligible studies in other languages will be excluded and that a proportion of the evidence base may be lost. The definition of eligibility criteria may have been overly rigid in the inclusion of the adolescent population only, which means that potentially useful findings from adult-based research will be excluded.

## Figures and Tables

**Table 1 brainsci-11-00764-t001:** Search strategy terms combined with the Boolean operator AND.

Keywords	Alternative Search Terms
Magnetic resonance imaging	(Neuroimaging[mh] OR neuroimaging[tiab] OR brain imaging*[tiab] OR brain scan*[tiab] OR resonance imaging[tiab] OR magnetic resonance imaging[mh] OR imaging technique*[tiab] OR brain/diagnostic imaging[mh] OR diffusion tensor imaging[mh] OR diffusion tensor imaging[tiab] OR DTI[tiab] OR susceptibility weighted imaging[tiab] OR SWI[tiab] OR MRI[tiab] OR fMRI[tiab] OR structural magnetic resonance imaging[tiab] OR sMRI[tiab])
Alcohol use	(alcohol drinking[mh] OR alcohol related disorders[mh] OR alcoholic beverages[mh] OR alcoholic beverage*[tiab] OR wine*[tiab] OR beer*[tiab] OR spirits[tiab] OR liquor*[tiab] OR (alcohol*[tiab] AND (drink*[tiab] OR beverage*[tiab] OR intoxicat*[tiab] OR abus*[tiab] OR misus*[tiab] OR risk*[tiab] OR consum*[tiab] OR excess*[tiab] OR problem*[tiab])) OR (drink*[tiab] AND (excess*[tiab] OR heavy[tiab] OR heavily[tiab] OR hazard*[tiab] OR binge[tiab] OR harmful[tiab] OR problem*[tiab])) OR alcohol use disorder*[tiab])
Adolescent	(child*[tiab] OR boy[tiab] OR boys[tiab] OR girl[tiab] girls[tiab] OR minor[mh] OR minors[tiab] OR adolescent[mh] OR adolescen*[tiab] OR teen[tiab] OR teens[tiab] OR teenager*[tiab] OR youth[tiab] OR youths[tiab] OR youngster*[tiab] OR young people[tiab] OR young person*[tiab] OR juvenile*[tiab] OR under ag*[tiab] OR underage*[tiab])

## Data Availability

Not applicable.

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
