# Peer review of "The Use of Magnetic Resonance Imaging Techniques in Assessing the Effects of Alcohol Consumption and Heavy Drinking on the Adolescent Brain: A Scoping Review Protocol"

_brainsci, 2021, doi:10.3390/brainsci11060764_

Round 1
Reviewer 1 Report
This study presents a scoping review protocol on the use of MRI in assessing alcohol consumption effects on the adolescent brain. Heavy drinking during adolescence has been shown to induce adverse structural brain changes. The aim of the study is quantifying and evaluating the quality of the studies in which MRI techniques are used to estimate regional brain deficits among heavy drinking adolescents.
The authors present a protocol following the guidelines in (Arksey and O’Malley 2005) scoping review methodology framework. They will present the results using Preferred Re-porting Items for Systematic Reviews and Meta-Analyses [PRISMA] extension for Scoping Reviews (PRISMA-ScR) guidelines. In order to assess the trustworthiness and the relevance of published papers, they use the critical appraisal tools provided by Joanna Briggs Institute.
The study relevant important in the sense where the effects of alcohol consumption on adolescents’ brain have been studied intensively but not many reviews have been published to highlight the findings especially that adolescence is a critical period for brain development. This study will summarize the scope of evidence and synthetize the findings of this problematic through the quantification and the evaluation of existing studies.
The paper is well written but it is not sufficiently substantial to be published. The study needs to be supplemented with the results. The authors should provide the statistics of the literature findings (number of papers, included and excluded papers…), present the extracted variables, perform the analysis and draw their conclusions. They can resubmit their paper then.
Additionally, there are some issues that need to be resolved.
- The paper is not very well organized. The authors should review the structure of the paper and improve it.
-The protocol stages are not very well highlighted. They are embedded in the sections’ paragraphs and are not very well highlighted. The authors should follow a logical structure like in () to facilitate the reading and the comprehension of the protocol.
- The sections titles are not very informative. Please improve and extend the titles so that they are more meaningful. (replace method with methodology, database as a title is not very informative…)
-The main question is stated very broadly. The authors should give more details and decompose it into sub-questions so that the reader understands all the steps of the study.
Minor issues
- The authors should not number the introduction section
-Section method Stage 2: Identifying and searching relevant studies: the authors should start by searching and then identifying
Author Response
7 May 2021
To: Prof Rebecca Xing
Editor: Brain Sciences
RE: Response to Reviewer 1 (Manuscript ID: brainsci-1179335)
The authors wish to thank Editor for the opportunity to resubmit and the reviewer for the very valuable comments and suggestions. We provide a point-by-point response to each comment (highlighted in yellow) below. We also indicate where revisions were made in the manuscript and attach the revised manuscript titled, “The use of magnetic resonance imaging techniques in assessing the effects of alcohol consumption and heavy drinking on the adolescent brain: a scoping review protocol”.
Please see the attachment
Sincere regards
Ms Nancy Hornsby
On behalf of co-authors

Reviewer 2 Report
The authors descripe the methodology they aim to employ for scoping review regarding the use of MRI for assessing the effects of alcohol consumptions in adolescence. The proposed methodology sounds good, nevertheless, I have easily found a more recent review from 2017 (https://www.frontiersin.org/articles/10.3389/fpsyg.2017.01111/full) about the use of MRI to study the effects of drinking in adolescence. The authors should acknowledge this work and consider if their proposed scoping review is still relevant given this more recent review. The authors should include this reference and adjust their motivation accordingly.
Moreover, they should address some minor issues in the following sections:
- No actual limitations are presented.
4.1 please state that terms will also be searched in the title and abstract, i.e. the [tiab] field code.
4.2.2 what if a study involves MRI and other imaging modalities? Will the MRI part be included? Please clarify.
4.2.2 systematic reviews are both in the inclusion and exclusion criteria.
- Authors state that they include grey literature (e.g. theses, conference papers), however, these are part of the exclusion criteria. Please clarify if these would be included or not.
Author Response
To: Prof Rebecca Xing
Editor: Brain Sciences
RE: Response to Reviewer 2 (Manuscript ID: brainsci-1179335)
The authors wish to thank Editor for the opportunity to resubmit and the reviewer for the valuable feedback and recommendations. We provide a point-by-point response to each comment (highlighted in yellow) below. We also indicate where revisions were made in the manuscript and attach the revised manuscript titled, “The use of magnetic resonance imaging techniques in assessing the effects of alcohol consumption and heavy drinking on the adolescent brain: a scoping review protocol”.
Please see the attachment.
Sincere regards

Round 2
Reviewer 1 Report
Thank you for the clarifications. The paper is much more improved.
Author Response
Thank you.